# The Moderated Mediation Effect of Age and Relaxation on the Relationship between the Type A Behavior Pattern and Blood Pressure in South Korea

**DOI:** 10.3390/healthcare11162264

**Published:** 2023-08-11

**Authors:** Sunghee An

**Affiliations:** Graduate School of Education, Hongik University, 94 Wausan-ro, Seoul 04066, Republic of Korea; ansh@hongik.ac.kr

**Keywords:** type A behavior pattern, blood pressure, relaxation, age, moderated mediation model

## Abstract

Although it is generally known that the type A behavior pattern (TABP) is associated with high blood pressure, recent studies have not consistently supported this association. Therefore, it is important to determine which factors moderate and mediate this relationship. The purpose of this study was to verify the moderated mediation effect of age and relaxation in the relationship between TABP and high blood pressure among middle-aged adults in South Korea. This study utilized data from the Korea Health and Genome Study (KHGS) supported by the Korea National Institute of Health. The participants in the present study were 10,022 Korean adults aged 40 to 69. To identify the conditional indirect effect of TABP on blood pressure through relaxation, with a consideration of age, a moderated mediation model using SPSS PROCESS macro 3.10 was applied. As is generally known, TABP was associated with high blood pressure, but its relevance was reversed through relaxation. Moreover, this relationship was moderated by age, which is known to be the most powerful predictor of high blood pressure. The lower age group with TABP exhibited a higher relaxation, whereas the higher age group with TABP demonstrated a lower relaxation. The findings demonstrate the need for relaxation education, stress management, or counseling to help individuals recognize their behavior patterns and strengthen the willingness and motivation for relaxation, especially among elderly people with TABP, in order to manage their blood pressure effectively.

## 1. Introduction

High blood pressure is major risk factor of cerebrovascular disorders and cardiovascular disease among middle-aged adults in South Korea [1], as well as worldwide [2]. The early detection and subsequent management of hypertension is critical, leading to the investigation and well-documented research of associated risk factors. Alongside genetic and behavioral factors, psychosocial stress has been considered as a possible risk factor, although the etiology of hypertension remains unclear. As a psychological risk factor for high blood pressure, a short-tempered personality known as the type A behavior pattern (TABP) has been notably observed since its introduction by cardiologist Ray Rosenman and Meyer Friedman in the Journal of the American Medical Association in 1959 [3].

TABP is typically characterized by individuals who exhibit excessive ambition, aggression, competitiveness, work-related drive, impatience, a need for control, a focus on quantity over quality, and a vulnerability to stress. It includes specific behaviors such as muscle tension, alertness, rapid and empathic vocal styles, and an accelerated pace of activities. Emotional responses associated with TABP encompass irritation, hostility, and an increased potential for anger [4]. The original definition of the TABP described it as a pathological behavior primarily composed of competitiveness, excessive drive, and an enhanced sense of time urgency. It later evolved into an epidemiological concept used to predict the occurrence of cardiovascular disease, including hypertension. The concept of TABP has continued to enjoy public appeal, and has remained a subject of contemporary health research as well. Moreover, TABP could be considered as an ongoing exposure to stress rather than acute stressors, as it is an individual’s personality that leads them to behave in certain ways. This personality trait could have a more probable association with persistent blood pressure elevations and the onset of hypertension [5].

As such, TABP is a type of personality and behavior pattern that is vulnerable to psychosocial stress. It can be considered that people with TABP are continuously exposed to stress, regardless of the type or intensity of stressful event itself. Since it is well known that psychologically stressful factors can influence the pathogenesis of various physical diseases [6], it can be expected that the relationship between TABP and hypertension should be significant. However, while this association appeared to be supported by findings from studies in the 1970s and 1980s, recent subsequent studies have not consistently found evidence of the relationship between TABP and coronary heart disease [7,8]. Similarly, previous research works associating stress with hypertension have produced mixed findings. Some studies have observed a positive association [9,10], but some have shown no correlation [11], and even a negative relationship [12]. There is little plausible explanation for these inconsistent results, even though stress-related variables have been evaluated as a possible risk factor in the etiology of hypertension.

Such inconsistency could result from a failure to consider mediating factors that may operate within the relationship between TABP and blood pressure. Since mediating factors are modifiable and can function as intervention strategies, understanding how these factors operate in the relationship between TABP and blood pressure becomes critical, as management is just as important as the early detection of hypertension. It is also necessary to consider moderating factors alongside mediating factors in order to fully comprehend the mechanisms that explain why certain individuals with a short-tempered disposition develop high blood pressure while others do not. This endeavor allows us to view TABP as a variable that is not easily changed, potentially possessing even more helpful characteristics for certain individuals. Therefore, this point allow to assume that the effects TABP on blood pressure may differ among individuals exhibiting TABP.

In this study, relaxation is considered as one of the key factors that mediate the relationship between TABP and blood pressure. Given that individuals with TABP generally exhibit a hard-driving lifestyle, characterized by elevated levels of aggression, easily aroused anger, a sense of time urgency, and a drive for competitive achievement, they are more likely experience a higher level of physical and psychological tension and stress in their daily life [4]. For this reasons, individuals with high levels of TABP may develop their own coping strategies and attempt to implement them to reduce tension and stress. Activities such as taking breaks, engaging in leisure activities, communication with supportive team members, exercising, and time management are all effective relaxation techniques. Several studies have reported that relaxation techniques, including light exercise, can be utilized to lower blood pressure [13,14,15,16].

Furthermore, most hypertension studies have reported that the effects of physical and behavioral factors, such as weight loss, diet, and exercise, on blood pressure vary across different age groups [17]. These factors demonstrated varying influences on blood pressure with age. Weight loss and exercise were more likely to be associated with decreased blood pressure in younger individuals compared to older individuals. Even adherence to hypertension medication exhibited variations across different age brackets [18]. Therefore, age should be considered as a moderating factor, because it is hypothesized that the effects of psychological and behavioral factors differentially influence blood pressure across various age groups. Based on these considerations, the following hypotheses were tested in both systolic and diastolic blood pressure in this study. Figure 1 presents the hypothetical representation of the moderated mediation model.

**H1:** *TABP is positively associated with blood pressure*.

**H2:** *The relationship between TABP and blood pressure is mediated by relaxation (indirect relationship)*.

**H3:** *The indirect relationship between TABP and blood pressure through relaxation is moderated by age (conditional indirect relationship)*.

## 2. Materials and Methods

### 2.1. Participants

This study utilized data from the Korea Health and Genome Study (KHGS), an ongoing population-based study of Korean adults aged 40–69, supported by the Korea National Institute of Health. The purpose of the KHGS is to investigate the frequencies and incidences of illnesses and their relationships with risk factors such as lifestyles, physical conditions, alcohol and food intake, smoking, psychological states, and personality traits. Trained interviewers and technicians conducted examinations through questionnaires, blood pressure measurements, anthropometries, and blood samplings. The data of 10,022 individuals from the 2008 samples were utilized in the present study.

### 2.2. Measures

The trained interviewers administrated all questionnaires, and technicians conducted blood pressure measurements. TABP was assessed using 10 items on a 4-point Likert scale ranging from 1 (not at all) to 4 (almost always). Example items include “People told me that I am easily upset” and “I can’t stand being late for an appointment or being slow to do something”. Relaxation was assessed with 3 items using response alternatives. An example item is “I try to relax by spending leisure time after work or on weekends”. The internal consistency reliabilities (Cronbach’s α) were 0.75 for TABP and 0.70 for relaxation.

Blood pressure (BP) was measured following the guidelines of the International Society of Hypertension of the World Health Organization (WHO-ISH), using a mercury sphygmomanometer operated by a trained technician. Caffeine intake and smoking were restricted for thirty minutes before the measurements. BP readings were taken three times on either arm with thirty-second intervals. The mean score of the three measurements from both arms was used in this study. Age was calculated based on the date of birth.

### 2.3. Data Analyses

The methodology of Gutiérrez-Cobo and colleagues [19] was adopted for data analysis. This study initially employed correlational analysis to identify the relationships among the research variables (i.e., TABP, relaxation, age, and blood pressure). Subsequently, a mediation analysis was conducted to identify the indirect effect of TABP on blood pressure through relaxation. In the third step, a moderation analysis was performed to examine the conditional effect of age on the relationship between TABP and relaxation. To identify the conditional indirect effect of TABP on blood pressure through relaxation, with age as a moderating factor, the final step involved applying the moderated mediation model. All analyses were carried out using SPSS PROCESS macro 3.10 [20,21]. Figure 1 illustrates the conceptual model of this research. The mediation effects were calculated using the bootstrapping procedure. The moderation effect was computed at three age values; 43 years old (16%: one SD below the average), 51 years old (50%: average), and 64 years old (84%: one SD above the average).

## 3. Results

### 3.1. Descriptive Statistics and Correlations

The comprehensive findings, as presented in Table 1, encompass an array of statistical measures including means, standard deviations, and correlation coefficients, effectively capturing the relationships among all research variables. Undoubtedly, age emerges as the most influential variable when examining the connections to both systolic and diastolic blood pressure. This established fact underscores the critical role that age plays in shaping these physiological indicators. Furthermore, it becomes evident that TABP exhibits a positive relationship with both systolic and diastolic blood pressure. This implies that as TABP increases, the levels of systolic and diastolic blood pressure also rise. Conversely, the relaxation variable demonstrates a negative correlation with systolic and diastolic blood pressure. This suggests that higher levels of relaxation tend to align with lower systolic and diastolic blood pressure readings.

### 3.2. Moderated Mediation Model Predicting Blood Pressure

The mediation, moderation, and moderated mediation analyses conducted for systolic and diastolic blood pressure are outlined in Table 2 and Table 3. The outcomes presented in Table 2 and Table 3 reveal that TABP was identified as a positive predictor for both systolic and diastolic blood pressure. Specifically, the coefficient for systolic blood pressure was 1.756 (95% CI [1.103, 2.409]), and for diastolic blood pressure, it was 0.733 (95% CI [0.327, 1.138]). These findings corroborate the first hypothesis, indicating that TABP exerts a positive influence on both types of blood pressure.

Moreover, the analysis revealed that TABP also functions as a positive predictor of relaxation, with a coefficient of 0.313 (95% CI [0.244, 0.382]). Conversely, relaxation was identified as a negative predictor of both systolic and diastolic blood pressure. The coefficient for systolic blood pressure was −6.719 (95% CI [−7.785, −5.653]), while for diastolic blood pressure, it was −2.400 (95% CI [−3.063, −1.738]). These results affirm the concept that higher levels of relaxation correspond to lower systolic and diastolic blood pressure. Subsequent mediation analysis unveiled a significant indirect effect of TABP on systolic and diastolic blood pressure, mediated through relaxation. The coefficient for the indirect effect on systolic blood pressure was −0.878 (95% CI [−1.040, −0.721]), and for diastolic blood pressure, it was −0.313 (95% CI [−0.404, −0.228]). These findings reinforce the second hypothesis, suggesting that individuals with a high TABP are more inclined to engage in relaxation, resulting in lower levels of both systolic and diastolic blood pressure.

Furthermore, the results of the moderation analysis revealed a significant interaction between TABP and relaxation. Moreover, the moderation effect of age was taken into consideration. To estimate the conditional effect of TABP on relaxation, the pick-a-point approach [20] was employed, considering three age values; 43 years old (average −1 SD), 51 years old (average), and 64 years old (average +1 SD). The findings demonstrated that the relationship between TABP and relaxation is more robust at 43 years old, with a coefficient of 0.159 (95% CI [0.142, 0.177]). This indicates that the influence of TABP on relaxation is more pronounced among individuals in this age group. In comparison, at 51 years old, the coefficient was 0.130 (95% CI [0.118, 0.142]), while at 64 years old, it was 0.084 (95% CI [0.065, 0.102]).

These results provide support for the third hypothesis proposed in this study, which suggests that the relationship between TABP and relaxation is influenced by age. Figure 2 visually presents a graphical representation of the moderation effect. The findings indicate that the association between TABP and relaxation varies depending on the individual’s age. The strongest relationship is observed at 43 years old, underscoring the significance of age as a moderating factor in comprehending the impact of TABP on relaxation.

After identifying the mediation and moderation effects, a moderated mediation analysis was conducted. The results reveal a significant conditional indirect effect on both systolic and diastolic blood pressure levels. For systolic blood pressure, the coefficient was 0.024 (95% CI [0.014, 0.034]), and for diastolic blood pressure, it was 0.009 (95% CI [0.005, 0.013]). These findings suggest that while TABP indirectly contributes to increased levels of systolic and diastolic blood pressure, the presence of relaxation as a mediating factor becomes particularly beneficial in reducing blood pressure in individuals with TABP, especially within the younger middle-aged adulthood range (40–50 years old). This underscores the importance of relaxation in mitigating the adverse effects of TABP on blood pressure, especially during the middle-aged adult years. By incorporating relaxation practices, individuals in this age group (40–50 years old) may experience a more significant reduction in systolic and diastolic blood pressure levels, thus promoting better cardiovascular health.

## 4. Discussion

### 4.1. Summary of Results

By addressing the inconsistent research outcomes regarding the relationship between TABP (type A behavior pattern) and blood pressure, this study aimed to further explore the potential roles of age and relaxation as moderating and mediating variables in this association. Age, acknowledged as one of the foremost influential factors in predicting high blood pressure [22], is believed to interact with non-pharmacological factors like dietary adjustments, exercise, stress reduction, or reduced alcohol consumption in effectively managing blood pressure [23]. Building upon prior research, this study placed specific emphasis on relaxation as a mediator, viewing it as a psychological variable that interacts with age and is closely linked with TABP.

Consistent with previous findings, this study reinforces the idea that TABP is correlated with elevated blood pressure levels [24]. Individuals exhibiting TABP traits, characterized by being time-consciousness and competitiveness, may be particularly prone to stress stemming from goal-driven behaviors. Notably, an extensive national sample unveiled a substantial link between heightened goal-driven stress and the prevalence of self-reported hypertension, with no observed moderating effects of race or ethnicity [25]. It is worth mentioning that race or ethnicity is recognized as another influential factor in relation to elevated blood pressure [26]. Hence, behavioral traits such as TABP might render individuals more susceptible to stress, potentially leading to unmanaged elevation in blood pressure levels if left unchecked.

Individuals with TABP are more susceptible to experiencing excessive stress, consequently heightening their need for relaxation. Interestingly, the outcomes of this study unveiled a positive association between TABP and relaxation, whereas relaxation was found to exhibit a negative correlation with blood pressure. Moreover, relaxation emerged as a significant mediator in the relationship between TABP and blood pressure. These findings potentially shed light on the inconsistent results observed in previous studies exploring the link between TABP and hypertension. It appears that certain modifiable factors, such as relaxation, hold the potential to influence the connection between TABP and blood pressure. In addition to relaxation, lifestyle adjustments, stress and depression management, and interventions addressing socioeconomic status (SES) have been acknowledged as effective psychosocial measures for reducing hypertension [27]. For individuals with TABP, effectively managing emotional stress through non-pharmacological approaches like exercise can contribute to the reduction in their blood pressure levels [28]. By incorporating these interventions, individuals with TABP can proactively engage in endeavors to manage their hypertension and foster enhanced cardiovascular well-being.

The results of this study provide support for the hypothesis concerning the moderating role of age. Specifically, the findings indicated a significantly stronger conditional indirect association between TABP, relaxation, and blood pressure among individuals aged 43, compared to those aged 51 and 64. This suggests that relatively younger middle-aged individuals with TABP are more prone to adopting relaxation practices than older individuals with TABP, ultimately resulting in blood pressure reduction. The moderated mediation model illuminated the critical significance of incorporating relaxation practices, particularly for individuals with TABP under the age of 60. Within this age group, a significant indirect relationship between TABP, relaxation, and blood pressure was established, underscoring the key role of relaxation in reducing blood pressure levels for these individuals. Conversely, no significant indirect relationship was observed between TABP and blood pressure among individuals aged 60 and above. These findings emphasize the varying impact of age on the relationship between TABP, relaxation, and blood pressure, suggesting that younger individuals with TABP may derive more substantial benefit from actively embracing relaxation practices to lower their blood pressure. On the other hand, among individuals aged 60 and above, additional factors or mechanisms may come into play, potentially influencing the connection between TABP and blood pressure.

The outcome of this study illuminates the underlying factors contributing to the divergent outcomes observed in prior research works concerning the link between TABP and blood pressure. It becomes apparent that the inclusion of diverse moderators and mediators, including age and psychological–behavioral variables, is pivotal in comprehending this intricate association. By incorporating these elements, researchers can attain a more holistic comprehension of how TABP impacts blood pressure and discern effective intervention approaches.

Mediating variables, like relaxation in this study, can serve as valuable intervention targets. The findings imply that advocating relaxation practices as a non-pharmacological means to reduce blood pressure could be especially advantageous for individuals under the age of 60 who have been diagnosed with hypertension and exhibit TABP traits. Offering education on comprehending one’s personality and behavioral type, coupled with highlighting the significance of relaxation, can empower individuals with essential knowledge and tools to proactively regulate their blood pressure levels.

By customizing interventions according to the distinct characteristics and requirements of individuals, healthcare professionals can enhance the efficacy of their strategies. Acknowledging age as a moderating variable, interventions can be precisely tailored to concentrate on and cater to the necessities of individuals within the younger age group (<60 years old) who are diagnosed with hypertension and manifest TABP traits.

### 4.2. Implications

The findings of this study hold significant practical implications for the formulation of clinical intervention and prevention programs aimed at individuals in their 40s and 50s exhibiting TABP. Considering the shielding role of relaxation in blood-pressure management, forthcoming initiatives should prioritize the integration of relaxation practices alongside established protective factors like a wholesome diet and consistent exercise.

Incorporating stress management initiatives into these interventions can yield considerable benefits. Approaches such as mindfulness-based stress reduction (MBSR) [29,30], acceptance and commitment therapy (ACT) [31,32], and autogenic training [33] present potential choices for inclusion in relaxation training programs. These methods have demonstrated effectiveness in alleviating stress and fostering relaxation, rendering them suitable for individuals with TABP.

Moreover, it is essential to acknowledge that relaxation training need not be exclusively delivered through professional programs. Encouraging individuals with TABP to consciously embrace relaxation in their daily routines can also yield a positive influence in warding off the onset of hypertension. Engaging in leisure pursuits, carving out dedicated moments of respite, or practicing uncomplicated relaxation practices can collectively contribute to stress reduction and effective blood-pressure management for individuals with TABP.

To summarize, the practical implications derived from this study advocate that the forthcoming clinical intervention and prevention initiatives should concentrate on integrating relaxation practices for individuals in their 40s and 50s who exhibit TABP. These programs can encompass stress management methodologies such as MBSR, ACT, and autogenic training. Additionally, promoting relaxation in everyday life through leisurely pursuits and regular intervals of relaxation can further fortify safeguards against hypertension development in individuals with TABP.

### 4.3. Limitations and Suggestions for the Future Studies

This study possesses several limitations that warrant acknowledgment. First, the use of cross-sectional data imposes constraints on the capacity to establish causal inferences concerning the relationship between relaxation and blood pressure. The findings derived from this study are founded on statistical assumptions and estimations, underscoring the necessity for further investigations employing longitudinal or panel data. Longitudinal studies would furnish more robust evidence regarding the causal interplay between relaxation and blood pressure, while also delving into how these dynamics may evolve across varying age groups over time.

Secondly, despite benefiting from a substantial sample size, this study specifically concentrated on middle-aged individuals in their 40s to 60s. Consequently, the generalizability of the findings to other age brackets might be curtailed. Subsequent research should endeavor to encompass a broader spectrum of age groups, encompassing individuals over 70 years of age, considering the increasing life expectancy trends. Analyzing the interrelation between TABP, relaxation, and blood pressure in older populations can yield valuable insights into potential age-linked distinctions within this association. Thirdly, it is crucial to acknowledge that this study relied on self-report measures. The self-reported responses in this study might exhibit exaggeration, respondents could feel reluctant to divulge private details, and various biases could impact the outcomes, including social desirability bias.

Additionally, it should be noted that this study does not account for potential confounding variables that could influence the relationship between TABP, relaxation, and blood pressure. Factors such as family history, lifestyle habits, medication use, comorbidities, socioeconomic status, and other diseases such as diabetes may impact the findings. Future research should aim to control for these confounders to attain a more accurate understanding of the relationship under investigation. Finally, it is essential to acknowledge the limitations of this study, primarily stemming from its reliance on secondary data. While this research endeavors to incorporate the current and pertinent data available, the 2008 data utilized in this study could inadvertently introduce findings that do not align with the latest developments in the field.

In conclusion, while this study has contributed to our understanding of the mediating role of relaxation in the relationship between TABP and blood pressure, it is vital to recognize the inherent limitations in the study design. Future research should employ longitudinal data, encompass a broader age spectrum, and account for potential confounding variables to enhance the validity and generalizability of the findings.

## Figures and Tables

**Figure 1 healthcare-11-02264-f001:**
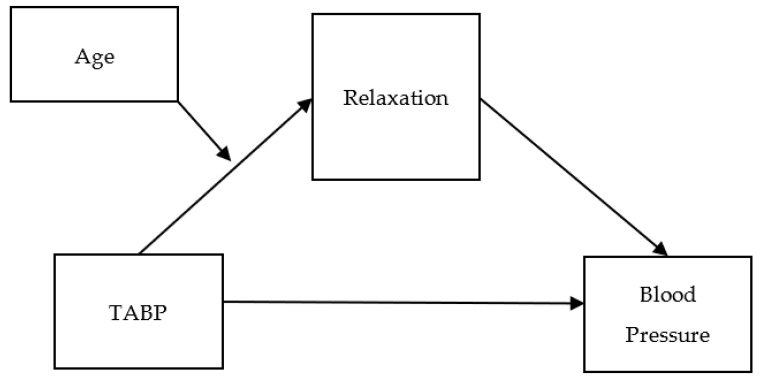
Hypothetical representation of the moderated mediation model.

**Figure 2 healthcare-11-02264-f002:**
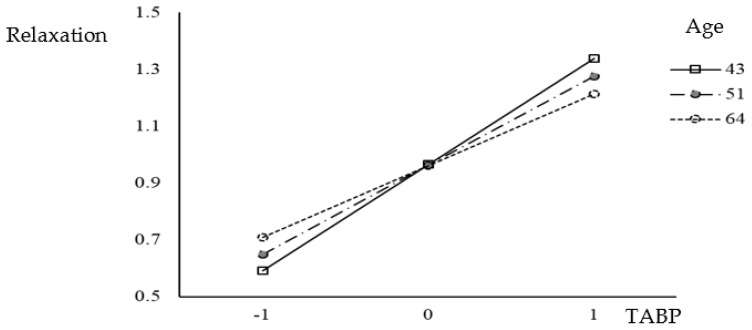
Moderation of the effect of TABP on relaxation by age.

**Table 1 healthcare-11-02264-t001:** Descriptive statistics and Pearson’s correlations among the study variables.

Variables					Pearson’s r			
	Mean	SD	Min.	Max	1	2	3	4
1. Age	52.29	8.93	40	69	―			
2. TABP	1.87	0.57	1	4	0.094 *	―		
3. Relaxation	0.31	0.35	0	2	−0.212 *	0.180 *	―	
4. Systolic BP	121.68	18.48	75	223	0.374 *	0.033 *	−0.116 *	―
5. Diastolic BP	80.33	11.44	34	155	0.207 *	0.026 *	−0.066 *	0.808 *

* *p* < 0.001.

**Table 2 healthcare-11-02264-t002:** Relationship between TABP and systolic blood pressure mediated by relaxation (N = 10,020). [SE = standard error; BCa CI = bootstrap accelerated confidence interval].

	B	SE	Bootstrapped SE	95% CI	95% BCa CI
Mediator model (DV = relaxation)					
TABP (predictor)	0.313 *	0.035		0.244~0.382	
Age (moderator)	−0.002	0.002		−0.004~0.001	
TABP × Age (interaction term)	−0.036 *	0.001		−0.049~−0.023	
Dependent model (DV = systolic BP)					
TABP (predictor)	1.756 *	0.333		1.103~2.409	
Relaxation (mediator)	−6.719 *	0.544		−7.785~−5.653	
Conditional indirect association by age					
43 years old	−1.071		0.105		−1.282~−0.867
51 years old	−0.878		0.080		−1.040~−0.721
64 years old	−0.566		0.074		−0.815~−0.425
Moderated mediation effect	0.024		0.005		0.014~0.034

* *p* < 0.01.

**Table 3 healthcare-11-02264-t003:** Relationship between TABP and diastolic blood pressure mediated by relaxation (N = 10,020). [SE = standard error; BCa CI = bootstrap accelerated confidence interval].

	B	SE	Bootstrapped SE	95% CI	95% BCa CI
Mediator model (DV = relaxation)					
TABP (predictor)	0.313 *	0.035		0.244~0.382	
Age (moderator)	−0.002	0.002		−0.004~0.001	
TABP × Age (interaction term)	−0.036 *	0.001		−0.049~−0.023	
Dependent model (DV = diastolic BP)					
TABP (predictor)	0.733 *	0.207		0.327~1.138	
Relaxation (mediator)	−2.400 *	0.338		−3.063~−1.738	
Conditional indirect association by age					
43 years old	−0.382		0.056		−0.499~−0.277
51 years old	−0.313		0.045		−0.404~−0.228
64 years old	−0.202		0.035		−0.277~−0.137
Moderated mediation effect	0.009		0.002		0.005~0.013

* *p* < 0.01.

## Data Availability

Publicly available datasets were analyzed in this study. This data can be found here: https://www.nih.go.kr/ko/main/contents.do?menuNo=300563 (accessed on 3 July 2023).

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
