# Peer review of "The Moderated Mediation Effect of Age and Relaxation on the Relationship between the Type A Behavior Pattern and Blood Pressure in South Korea"

_healthcare, 2023, doi:10.3390/healthcare11162264_

Round 1
Reviewer 1 Report
The authors present a well-designed and adequately conducted large-scale cross-sectional study on the relationship between TABP and blood pressure, studying the moderating factors along with mediating factors to understand the underlying mechanisms fully. Applying advanced statistics methods (SPSS PROCES macro), they revealed a significant interaction between TABP and relaxation, being further moderated by age. The study proved a significant conditional indirect effect of relaxation on both systolic and diastolic blood pressure levels, especially in middle-aged (40 – 50 years) adults, promoting better cardiovascular health.
In a practical sense, the important finding is that also they have found that relaxation training was not necessarily delivered through professional programs, but that the individual efforts towards relaxation including leisure activities were effective as well.
The authors adequately defined study limitations (cross-sectional type, design based on statistical assumptions and estimates) pointing out the need for further longitudinal studies to assess the putative causal relationship between relaxation and blood pressure, and the related differences among different age groups.
Overall, the study is well designed and properly conducted, the article is well presented, with sufficient literature support, and with conclusions based on the achieved findings. English language and style are fine. This article merits publication in the given form.
Author Response
Thank you so much for your compliment.
Reviewer’s comment: The authors present a well-designed and adequately conducted large-scale cross-sectional study on the relationship between TABP and blood pressure, studying the moderating factors along with mediating factors to understand the underlying mechanisms fully. Applying advanced statistics methods (SPSS PROCES macro), they revealed a significant interaction between TABP and relaxation, being further moderated by age. The study proved a significant conditional indirect effect of relaxation on both systolic and diastolic blood pressure levels, especially in middle-aged (40 – 50 years) adults, promoting better cardiovascular health.
Author’s response: Thank you so much for your compliment.
Reviewer’s comment: In a practical sense, the important finding is that also they have found that relaxation training was not necessarily delivered through professional programs, but that the individual efforts towards relaxation including leisure activities were effective as well.
Author’s response: Yes, I agree with your opinion.
Reviewer’s comment: The authors adequately defined study limitations (cross-sectional type, design based on statistical assumptions and estimates) pointing out the need for further longitudinal studies to assess the putative causal relationship between relaxation and blood pressure, and the related differences among different age groups. Overall, the study is well designed and properly conducted, the article is well presented, with sufficient literature support, and with conclusions based on the achieved findings. English language and style are fine. This article merits publication in the given form.
Author’s response: Thank you so much for your compliment.
Reviewer 2 Report
In this study, authors tried to demonstrate the correlation of TABP and blood pressure, especially if relaxation influences this correlation. However, several key points were left unexamined, and some important details are not described. Here are my major concerns:
1, How did authors quantify the "relaxation"? There are so many different types of "relaxation", and also the duration varies a lot. Also, the same relaxation is highly likely to have significantly different effects on different individuals. Thus, authors need to provide more details regarding how they quantify the relaxation.
2, Authors should at least compare the blood pressure between individuals with more and less relaxation groups.
3, Authors did not list the prevalence of other baseline diseases, which are important causes of elevated blood pressure.
4, Substantial work will be needed to improve the writing.
English writing needs to be substantially improved.
Author Response
Reviewer’s comment:
In this study, authors tried to demonstrate the correlation of TABP and blood pressure, especially if relaxation influences this correlation. However, several key points were left unexamined, and some important details are not described. Here are my major concerns:
Author’s response: Thank you so much for your valuable comments. I will carefully response your comments one by one.
1, How did authors quantify the "relaxation"? There are so many different types of "relaxation", and also the duration varies a lot. Also, the same relaxation is highly likely to have significantly different effects on different individuals. Thus, authors need to provide more details regarding how they quantify the relaxation.
Author’s response: As I stated in the method section, I utilized the data from Korean Health and Genome Study (KHGS). In KHGS, relaxation variable was measured with following 3 items.
- I try to relax my tension having leisure time after work or on weekends.
- It is hard to find time to relax (reverse item).
- In order to relax myself, sometimes I used to take prescribed medication.
Additionally, I addressed the limitation for self-reported measurement in limitation section.
In text (p.9): Thirdly, this study used self-report measures. Self-reported answers in this study may be exaggerated; respondents may be too embarrassed to reveal private details; various biases may affect the results, like social desirability bias.
2, Authors should at least compare the blood pressure between individuals with more and less relaxation groups.
Author’s response: Thank you so much for your valuable comments. Relaxation variable was continuous variable. Therefore, I reported the correlation coefficient value between relaxation and blood pressure. As expected, the relationship was negative. I reported this results in the results section as follow.
In text (p.4): the variable of relaxation demonstrates a negative correlation with systolic and diastolic blood pressure. This suggests that higher levels of relaxation tend to coincide with lower systolic and diastolic blood pressure readings.
3, Authors did not list the prevalence of other baseline diseases, which are important causes of elevated blood pressure.
Author’s response: Yes, I agree with your opinion. I believe that there are other important causes of elevated blood pressure. I list those issues in introduction and limitation sections.
In text (p.1): Along with genetic and behavioral factors, psychosocial stress has been accounted as possible risk factor though the etiology of hypertension still not clear.
In text (p.9): Factors such as family history, lifestyle habits, medication use, comorbidities, and socioeconomic status could have an impact on the findings.
4, Substantial work will be needed to improve the writing.
Author’s response: I asked the native speaker to proofread the revised version of manuscript. Thank you for your valuable comments.
Reviewer 3 Report
Title: The Moderated Mediation Effect of Age and Relaxation on the Relationship between Type A Behavior Pattern and Blood Pressure in South Korea.
Reviewer Comments: Authors conducted this study to identify the mediation effect of age and relaxation in the relationship between TABP and blood pressure among middle aged people. Authors obtained data from the Korea Health and Genome Study. Participants were about 10,000 people aged 40 to 69. To explore conditional indirect effect, the moderated mediation model using SPSS PROCESS was applied. Authors identified that TABP was associated with high blood pressure, but its relevance can be reversed through relaxation. People in the lower age group with TABP had higher relaxation, on the other hand, people in the higher age group with TABP had lower relaxation.
Strengths:
1. Sample size.
2. These kinds of studies might improve people health, and cost efficient. We just need to educate them about the advantages of relaxation and ho its affect their health.
Weaknesses:
1. Only the age group from 40-69 included in the study. Other age groups also need to be included.
2. Panel data or longitudinal data is required.
3. People included in the study might have other diseases and they may be using medications. This might affect the results.
4. Other variables such as socioeconomic status and comorbidities might affect the results.
5. Data obtained from secondary sources, and it’s based on inferences of statistics.
Author Response
Reviewer Comments.
Authors conducted this study to identify the mediation effect of age and relaxation in the relationship between TABP and blood pressure among middle aged people. Authors obtained data from the Korea Health and Genome Study. Participants were about 10,000 people aged 40 to 69. To explore conditional indirect effect, the moderated mediation model using SPSS PROCESS was applied. Authors identified that TABP was associated with high blood pressure, but its relevance can be reversed through relaxation. People in the lower age group with TABP had higher relaxation, on the other hand, people in the higher age group with TABP had lower relaxation.
Strengths:
- Sample size.
- These kinds of studies might improve people health, and cost efficient. We just need to educate them about the advantages of relaxation and ho its affect their health.
Author’s response: Thank you so much for pointing out the strengths of my research.
Weaknesses:
- Only the age group from 40-69 included in the study. Other age groups also need to be included.
Author’s response: Yes, I agree with your opinion. The data is limited with 70 and above. Therefore, I address this issue in the limitation section.
In text (p.9): Second, while this study benefits from a large sample size, it focused specifically on middle-aged individuals in their 40s to 60s with TABP. Therefore, the generalizability of the findings to other age groups may be limited. Future research should aim to include a broader range of age groups, including individuals over the age of 70, as life expectancy has been increasing in recent years.
- Panel data or longitudinal data is required.
Author’s response: Yes, I agree with your opinion. Future study needs to extend with panel or longitudinal data. I address this issue in the limitation section.
In text (p.8): This study has several limitations that should be acknowledged. First, the utilization of cross-sectional data restricts the ability to make causal inferences regarding the relationship between relaxation and blood pressure. The findings of this study are based on statistical assumptions and estimates, and further research employing panel or longitudinal data is needed.
- People included in the study might have other diseases and they may be using medications. This might affect the results.
Author’s response: Yes, I agree with your opinion. Other diseases such as diabetes could be confounding variables. I address this issue in the limitation section.
In text (p.9): Factors such as family history, lifestyle habits, medication use, comorbidities, socioeconomic status, and other diseases such as diabetes could have an impact on the findings.
- Other variables such as socioeconomic status and comorbidities might affect the results.
Author’s response: Yes, I agree with your opinion. I address this issue in the limitation section.
In text (p.9): Factors such as family history, lifestyle habits, medication use, comorbidities, socioeconomic status, and other diseases such as diabetes could have an impact on the findings.
- Data obtained from secondary sources, and it’s based on inferences of statistics.
Author’s response: Yes, I agree with your opinion. I address this issue in the limitation section. Thank you for your valuable comments.
In text (p.9): Finally, it is essential to acknowledge the limitations of this study, primarily based on secondary data. While this research aims to incorporate the most current and relevant data available, the 2008 data in this study might inadvertently introduce findings that do not reflect the most recent developments in the field.
Round 2
Reviewer 2 Report
Authors handled my questions well. But I still strongly suggest a through language editing.
A deep and through English editing will be needed.
Author Response
Thank you for your comments. I have revised the 2nd round manuscript thoroughly with the assistance of a native speaker of English.